

# A lightweight coal-gangue detection model based on parallel deep residual networks

Shexiang Jiang[1,2,*] and Xinrui Zhou[1,2,*]

[1] Anhui Key Laboratory of Mine Intelligent Equipment and Technology, Anhui University of Science and Technology, Huainan, Anhui, China
[2] School of Computer Science and Engineering, Anhui University of Science and Technology, Huainan, Anhui, China
* These authors contributed equally to this work.

## ABSTRACT

To realize the accurate identification of coal-gangue in the process of underground coal transportation and the low-cost deployment of the model, a lightweight coal-gangue detection model based on the parallel depth residual network, called P-RNet, is proposed. For the problem of images of coal-gangue taken under complex conditions, the feature extraction module (FEM) is designed using decoupling training and inference methods. Furthermore, for the problem of the nearest neighbor interpolation upsampling method being prone to produce mosaic blocks and edge jagged edges, a lightweight upsampling operator is used to optimize the feature fusion module (FFM). Finally, to solve the problem, the stochastic gradient descent algorithm is prone to local suboptimal solutions and saddle point problems in the error function optimization process, numerous experiments are carried out on selecting the initial learning rate, and the Lookahead optimizer is used to optimize parameters during backpropagation. Experimental results show that the proposed model can effectively improve the recognition effect, with a corresponding low deployment cost.

# INTRODUCTION

Coal preparation is an indispensable stage of coal production and an important part of clean coal technology, which is considered the most effective and economically valuable technology to reduce coal pollution to the environment (*Wang, Wang & Li, 2023*; *Xue et al., 2023a*; *Zhao et al., 2022*). Manual sorting is the main method for pre-sorting coal gangue, but it is inefficient and has many safety hazards. Mechanical sorting methods, such as heavy media and flotation, pose pollution risks, adversely impacting the environment. Automated sensor-based beneficiation technology can reduce downstream costs and improve ore quality, but these methods are susceptible to interference and have poor stability. Coal-gangue detection usually faces challenges, such as the target image being susceptible to light, coal dust, and motion blurs, making it difficult to detect (*Zhang et al., 2024*; *Zou et al., 2020*).

The method based on image processing has been widely used in the coal preparation field, and it has witnessed remarkable progress in recent years (*Wang et al., 2022b*). *Dou*

Corresponding author
Shexiang Jiang,
sxjiang8888@163.com

*et al. (2019)* employed color and texture feature extraction for gangue images, introducing a relief-SVM based on image analysis. However, their research focused solely on four coal properties, neglecting the complexity of multiple situations and imposing application limitations. Similarly, *Hu et al. (2019)* studied the detection method of coal-gangue, which is based on multispectral imaging and LBP feature extraction algorithm. Nevertheless, this work only considers the classification of coal and coal-gangue under the strategy of combining feature extraction and classifier, but different feature extraction methods and classifiers will have a certain impact on the results. *He et al. (2022)* proposed a concave point detection and segmentation method, adapted for multi-scale X-ray images of coal-gangue, but encountered implementation complexities. *Li et al. (2022)* used binarized and morphologically processed to obtain complete and clean gangue samples and analyzed these samples by using morphological corrosion and expansion methods. Nevertheless, the classification accuracy is lower. In fact, the implementation process of the method based on image processing is complex, and the application scenarios are limited. Furthermore, it is difficult to realize the further separation operation of coal-gangue only by studying the identification of coal and coal-gangue.

Recently, using machine learning to sort coal-gangue plays an important role in the coal field, which has been a research hotspot (*Lv et al., 2021*; *McCoy & Auret, 2019*; *Yan et al., 2022*; *Si et al., 2023*). With the rise of deep learning, many detectors based on deep neural networks (DNNs) have been proposed (*Yan et al., 2022*; *Lai et al., 2022*). To address the challenges posed by gangue images under multi-scale and semi-occlusion conditions, *Wang et al. (2022a)* introduced a semantic segmentation network based on a pyramid scene interpretation network. However, the method's training process exhibited considerable loss, as indicated by experimental results. For fine-grained characteristics, *Xue et al. (2023b)* studied the feature scaling and unstructured pruning of the model, and proposed a gangue detection algorithm, called ResNet18-YOLO. However, this method has not been tested in real environments, it is necessary to transplant it to the gangue sorting robot to carry out the verification of the actual gangue detection effect. *Zhang et al. (2022)* used the YOLOv4 algorithm based on deep learning for the detection of gangue, however, the experiments were carried out in an ideal environment, and the influence of potential impurities in the actual production environment was not considered, which led to the poor robustness of the model. *Lai et al. (2022)* proposed an improved mask R-CNN combined with the multispectral image gangue instance segmentation method, which lightened the classical mask R-CNN model. However, the details were not described in the experiments.

To address the above challenges effectively, a novel parallel deep residual network (P-RNet) is specifically designed and implemented for coal-gangue detection. Inspired by the residual learning principles of ResNet, P-RNet introduces an innovative component, the ParallelBlock, which enhances feature extraction through parallel residual paths, distinguishing it from standard ResNet architectures. This design not only achieves high recognition accuracy but also maintains low computational and deployment cost by incorporating lightweight techniques. The main contributions of this work are as follows:

- A novel parallel deep residual block, named ParallelBlock, is proposed as the backbone for the feature extraction module (FEM). Furthermore, a decoupled training and inference method is applied to enhance efficiency in this module.
- A custom lightweight upsampling operator is introduced in the feature fusion module (FFM) to amplify high-level features. This optimization significantly boosts the model's performance while keeping computational overhead minimal.
- The Lookahead optimizer is used for parameter learning during backpropagation. Extensive experiments were conducted to determine the optimal initial learning rate, ensuring stable and effective training.

The rest of the article is organized as follows. The proposed P-RNet is detailed in "Methodology". The experimental results and comparative analysis are illustrated in "Experiments". Finally, conclusions and prospects are drawn in "Conclusion and Future Work".

## METHODOLOGY

### Overview

Inspired by deep learning methods (*e.g.*, YOLO, ResNet, MobileNet, *etc.*) (*Sandler et al., 2018*; *He et al., 2015*), a lightweight coal-gangue detection model named P-RNet is designed, which is based on a parallel deep residual network. The overall architecture of P-RNet is illustrated in Fig. 1, and it mainly consists of four parts: Input, Backbone, Head, and Output.

First, image enhancement, data augmentation, and letterbox are used to reprocess the collected coal-gangue images. Next, the FEM serves as the backbone network to extract features from the coal-gangue images. Feature pyramid networks (FPN) (*Lin et al., 2017*) and path aggregation networks (PAN) (*Liu et al., 2018*) are used in the FFM. Specifically, the FPN structure uses lightweight operators for upsampling from the top-down; the PAN structure performs downsampling from the bottom-up. Finally, CIoU is introduced to calculate loss, and the eventual result is obtained.

### Feature extraction module

Actually, images of coal-gangue are taken under complex conditions, affected by many factors, such as light and coal-dusty (*Zhang et al., 2021*). Existing methods do not meet the requirements of practical applications. To extract more goal-gangue features, the ParallelBlock module is proposed in this article for resource-constrained equipment.

The deeper the network, the more information and richer features can be obtained (*Samek et al., 2021*; *Li et al., 2021*; *Simonyan & Zisserman, 2015*). However, experiments have shown that as the network deepens, accuracy tends to saturate and then degrade rapidly. Additionally, deeper networks are prone to gradient vanishing and exploding problems. Consequently, adding more layers beyond an optimal depth increases training errors (*He et al., 2015*; *He & Sun, 2015*).

To address the degradation problem, the module based on a deep residual learning framework is introduced, and stacked layers are used to perform the residual mapping.

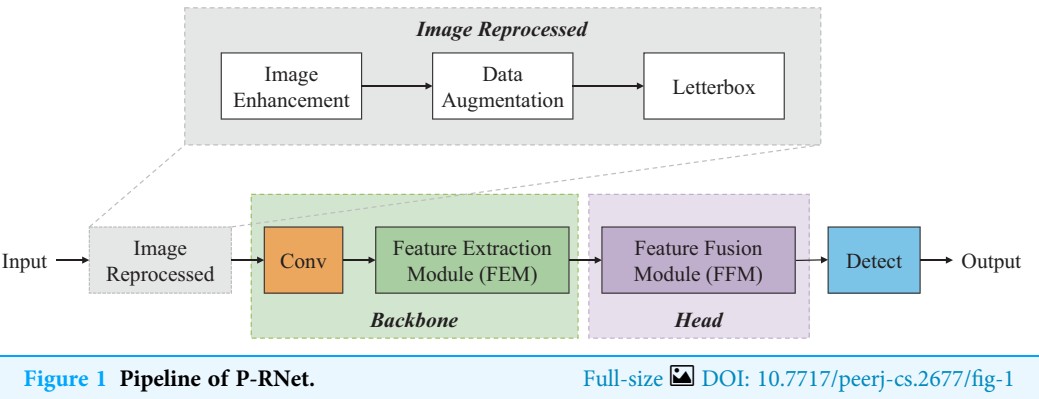

**Figure 1 Pipeline of P-RNet.**

However, many residual calculations will lead to increased model complexity. To this end, in this article, expansion convolution is introduced for dimensionality ascension first, and then depthwise convolution is used for feature extraction to obtain more information. Finally, projection convolution is applied for dimensionality reduction. Note that in depthwise convolution, the kernel only performs feature extraction on one channel, which is less computational.

Additionally, the training and inference process is designed, respectively. Specifically, high-precision multi-branch topology is used for weight learning during training, and a low-latency single-branch topology is used during inference. Finally, the weights of the multi-branch topology are transferred to the single-branch topology. Accordingly, the proposed module reaches a balance between speed and performance. For the specific task of coal-gangue detection, two ParallelBlock structures are proposed. Schematic visualization of different blocks is shown in Fig. 2.

In this article, y can be implemented by a feedforward neural network (*Svozil, Kvasnicka & Pospichal, 1997*; *Wei et al., 2018*). Since the identity mapping operation can add the input of a convolution operation directly to the output, it does not add additional parameters and computational complexity (*He et al., 2015*, *2016*). In summary, the ParallelBlock structure designed and implemented in this article is suitable for the coal-gangue detection model, and the corresponding deployment cost of the model is low.

Note that identity mapping can make training easier, but in real cases, it may not be optimal. In this study, we address this problem by reformulating.

Generally, as a backbone module, the FEM has several advantages. (1) Using identity mapping gives the model a multi-branch structure, facilitating multi-scale fusion. (2) The parallel structure is beneficial in conveying context information, which improves the performance of the model.

### Feature fusion module

To better utilize the features of coal-gangue extracted by the backbone network and make predictions, the FPN+PAN structure is used in the FFM. In the FPN structure, a lightweight upsampling operator Content-Aware ReAssembly of Features (CARAFE) (*Wang et al., 2019*), which can guide the reassembly process according to the input characteristics used. The structure of the FFM is shown in Fig. 3.

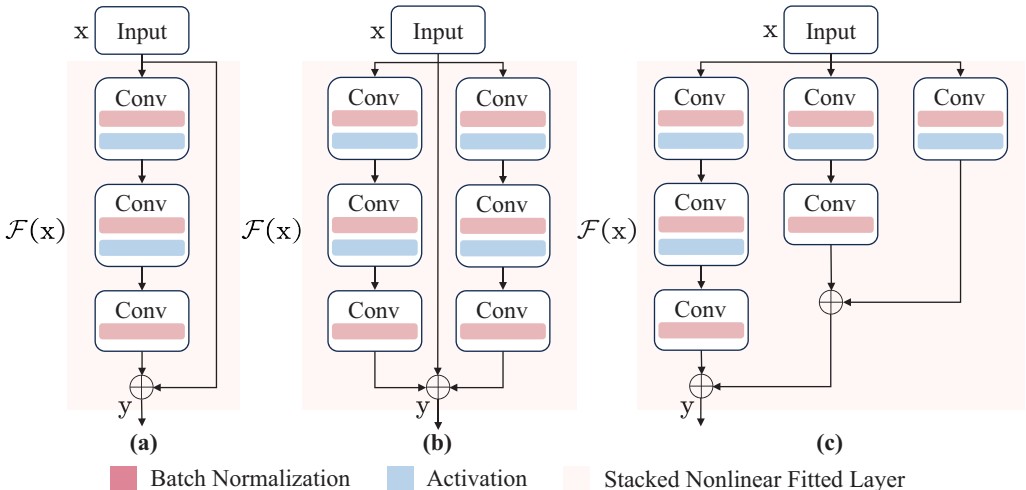

**Figure 2 The network architectures of inverted residual block and ParallelBlock.** (A) The inverted residual block. (B) and (C) the ParallelBlock. The function $\mathscr{F}(x)$ represents the stacked nonlinear fitted layer, expressed as $\mathscr{F}(x) = \mathscr{H}(x) - x$, where H(x) is the underlying mapping. Ultimately, the original mapping is reshaped to $y = \mathscr{F}(x) + x$, where y is output.

Specifically, the design of the FPN structure adopts a top-down architecture, which builds high-level semantic feature mappings at all scales. This structure transfers and fuses high-level information through upsampling to obtain predictive feature maps. In the PAN structure, a bottom-up architecture enhances the feature hierarchy by utilizing precise signals at lower levels. Furthermore, a fully connected operation connects the feature mesh and all feature layers, allowing semantic information to propagate directly to the subnetwork.

CARAFE comprises two key components: the upsampling prediction module and the feature reassembly module. To reduce computation, a $1 \times 1$ convolutional layer is used to compress the feature map channel. Then, a convolutional layer with kernel size $k_{encoder}$ is used to predict the upsampled kernel. The softmax function is used to normalize the reorganized kernel of $k_{up} \times k_{up}$, where $k_{up}$ is the size of the recombinant kernel, $k_{encoder} = k_{up} - 2$.

In the feature reassembly module, the features in the local area are reassembled using the weighted sum operator $\phi$, and the positions in the output feature map are mapped back to the input. Further, a dot product between the area of $k_{up} \times k_{up}$ centered on this position and the predicted upsampling kernel to obtain the output.

Intuitively, FFM has several advantages. (1) FPN+PAN structure conveys one low-level localization feature while passing high-level semantic features, which is beneficial to utilize the feature. (2) Due to the application of CARAFE, it can significantly improve the model's performance while obtaining a small amount of computation.

## Dynamic adjustment strategy of learning rate

Learning rate is one of the most critical hyperparameters for training deep neural networks, which can control the amplitude of each change of the parameter vector of deep neural networks. Moreover, it affects the performance of neural network classifiers to a

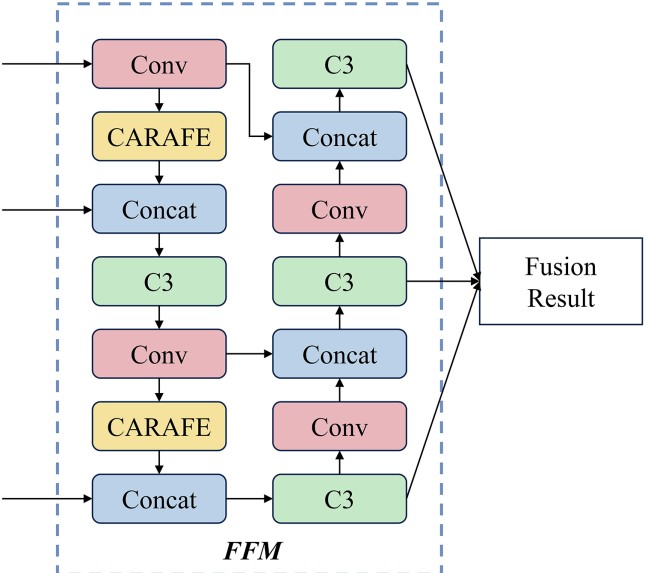

**Figure 3 The structure of feature fusion module.**

certain extent (*Smith, 2017*; *Breuel, 2015*). An oversized learning rate will cause the weight update to fail, and the loss function will produce oscillations or even fail to converge. When the learning rate is too small, the convergence of the loss function will be slow. In particular, the network parameters may fall into the saddle point, or the local minimum value cannot be updated, limiting neural network performance improvement.

To this end, an optimization algorithm is used to adjust the learning rate dynamically. In deep neural networks, the optimizer is primarily responsible for updating the model parameters to approach or reach the optimal value. Stochastic gradient descent (SGD) is the most commonly used optimizer. However, this optimizer can cause the model to easily fall into numerous local suboptimal solutions and saddle points during the error function optimization process. Moreover, SGD performs frequent updates with high variance, often requiring expensive hyperparameter tuning to achieve better performance in the neural network (*Ruder, 2017*).

To effectively address these issues and further improve the performance of P-RNet, the Lookahead (*Zhang et al., 2019*) optimizer is used in this study for hyperparameter optimization. First, copy the model parameters into two copies, defined as fast weights $\theta$ and slow weights $\phi$. Second, standard optimization algorithms (*e.g.*, SGD and Adam) and linear interpolation are used for updating fast and slow weights, respectively. Finally, after each $\phi$ update, $\theta$ is reset to the current $\phi$ value. Pseudocode for Lookahead is provided in Algorithm 1.

The Lookahead optimizer is less sensitive to suboptimal hyperparameters and can reduce the need for extensive hyperparameter tuning. Its use improves generalization while maintaining robustness to hyperparameter updates.

| **Algorithm 1**    **Lookahead optimizer.** |
| :--- |

**Input:** parameters $\phi_0$, objective function $L$, dataset $D$, synchronization period $k$, slow weights learning rate $\alpha$, optimizer $A$ (which is Adam)

**Output:** $\phi$

1:    **for** $t=1,2,\ldots$ **do**
2:       Synchronize parameters $\theta_{t,0} \leftarrow \phi_{t-1}$
3:       **for** $i=1,2,\ldots,k$ **do**
4:          sample mini-batch of data d $\sim D$
5:       **end for**
6:       Perform update $\theta_{i,t} \leftarrow \theta_{t,i-1} + A(L, \theta_{t,i-1}, d)$
7:    **end for**
8:    **return** $\varphi$

# EXPERIMENTS

## Settings

This study uses the PyTorch framework to design a neural network model, implemented in Python 3.9. The experimental environment includes a Windows 10 64-bit operating system, an Intel Xeon E5-2670 @ 2.60 GHz processor, 32 GB of RAM, and an NVIDIA GeForce RTX 2080Ti GPU with 12 GB of memory. The model parameters are configured with a depth of 0.33, a width of 0.50, a learning rate momentum of 0.937, 500 epochs, a batch size of 16, and the cross-entropy loss function for error evaluation.

## Coal-gangue dataset

In this article, coal–gangue images were collected from surveillance video at a coal preparation plant in Anhui Province, China. The training and validation dataset consists of 1,500 conveyor belt coal transport images. Due to issues with the acquired images being susceptible to light, coal dust, and blurring caused by the fast running speed of the conveyor, linear transformation and deblurring algorithms are used to preprocess the data. Data augmentation techniques are employed to improve the generalization ability of the model and enhance the robustness of the algorithm. A total of 1,000 images are used as training samples, and 430 images are randomly selected as test samples.

## Evaluation metrics

In this work, positive and negative samples will be used to label coal-gangue and non-gangue targets, respectively, and the classification is shown in Table 1.

There are several commonly used quantitative metrics for evaluating an object detection model. In this article, to assess the recognition performance of the proposed model, mean Average Precision (mAP), Precision, and Recall were selected as the evaluation metrics. These can be defined as follows:

**Table 1 Confusion matrix for classifying coal-gangue and non-gangue samples.**

| Forecast category | Actual category | |
| --- | --- | --- |
| | Coal-gangue | Non-gangue |
| Coal-gangue | True Positive (TP) | False Positive (FP) |
| Non-gangue | False Negative (FN) | True Negative (TN) |

$$\text{mAP} = \frac{1}{Q}\sum_{q=1}^{Q}\sum_{k=1}^{N} p(k) \times \Delta r(k) \tag{1}$$

$$\text{Precision} = \frac{TP}{TP + FP} \tag{2}$$

$$\text{Recall} = \frac{TP}{TP + FN} \tag{3}$$

## Learning rate analysis

In this work, the Lookahead optimizer is used to dynamically adjust the learning rate, where the initial learning rate (lr0) significantly affects the model's performance. As shown in Table 2, comparison experiments using both the SGD and Lookahead optimizers with different lr0 values were conducted. The results indicate that the values obtained with the Lookahead optimizer are more stable than those with SGD. Both optimizers achieve the best performance when lr0 = 0.007.

In addition, to more intuitively illustrate the impact of the optimizer on model performance, the training curves are shown in Fig. 4. The results indicate that the curve begins to converge around 50 epochs when using the Lookahead optimizer, demonstrating a faster convergence speed compared to the SGD optimizer. As shown in Figs. 4A, 4B and 4D, the convergence rate increases with higher lr0 values, but stability gradually deteriorates. Figure 4C shows that when lr0 is 0.003, 0.005, 0.007, or 0.009, the curves initially rise and then fall, indicating serious overfitting. When lr0 is set to 0.001, the curve rises stably as it converges with no overfitting observed.

In summary, the Lookahead optimizer achieves better generalization results than the SGD optimizer and converges faster. In addition, the Lookahead optimizer is more stable and robust during training. The Lookahead optimizer is used in this article to adjust the learning rate dynamically. The lr0 of subsequent experiments is set to 0.001 to prevent overfitting during training.

## Ablation study

Our proposed detector has two components: FEM and FFM. In order to analyze the effectiveness of each part, an ablation study is conducted in Table 3, and visualization and quantitative are carried out subsequently.

**Table 2 Experimental results between SGD and Lookahead with different lr0.**

| Optimizer | lr0 | mAP_0.5↑ | mAP_0.5:0.95↑ |
|---|---|---|---|
| SGD | 0.001 | 0.9950 | 0.5712 |
| | 0.003 | 0.9950 | 0.6092 |
| | 0.005 | 0.9950 | 0.6764 |
| | 0.007 | 0.9950 | 0.6781 |
| | 0.009 | 0.9778 | 0.6333 |
| Lookahead | 0.001 | 0.9950 | 0.6780 |
| | 0.003 | 0.9950 | 0.6561 |
| | 0.005 | 0.9950 | 0.6808 |
| | 0.007 | 0.9950 | 0.6824 |
| | 0.009 | 0.9950 | 0.6419 |

### Feature extraction module

To verify the effect of the ParallelBlock on the proposed detector, experiments were conducted using both the InvertedBlock and ParallelBlock in the backbone. The experimental results were analyzed using four indicators (*i.e.*, mAP_0.5, mAP_0.5:0.95, obj loss, and box loss). Figure 5 compares the performance of the detector with different backbones. As shown in Fig. 5, the curve begins to converge around 100 epochs.

Further analysis of Fig. 5 reveals that the two ParallelBlock structures have different effects on the detector. ParallelBlock_1 combines inverse residuals, convolution, and identity mapping in parallel at different locations. As seen in Fig. 5B, the curve for the model using this structure rises steadily and achieves higher values compared to models using other structures. ParallelBloc_2 combines double inverted residuals and identity mapping in parallel. The detector using this structure exhibits advantages in terms of box loss and object loss (Figs. 5C and 5D).

### Feature fusion module

The feature fusion with different upsampling methods (*i.e.*, nearest and CARAFE) is tested to evaluate the influence of the FFM, and the loss curves are shown in Fig. 6. According to the visualization results, as lr0 increases, the degree of model loss gradually decreases, with the curves beginning to converge around 100 epochs. Figures 6A and 6B show that the object loss initially increases, then gradually decreases after reaching a peak. In the five sets of experiments with different lr0, the loss degree with the CARAFE operator is lower than that with the nearest neighbor. Figures 6C and 6D indicate that in all five groups of experiments, the object loss shows a downward trend as the number of epochs increases.

To further demonstrate the effectiveness of the proposed detector, lr0 is set to 0.001 (Learning Rate Analysis), and the results obtained with different upsampling methods are shown in Fig. 7. It can be observed that the detector using CARAFE for upsampling shows a significant increase in mAP_0.5 and mAP_0.5:0.95 compared to the nearest neighbor method (Figs. 7A and 7B). As shown in Figs. 7C and 7D, when using CARAFE, the Precision and Recall curves begin to converge and stabilize around 100 epochs. In contrast,

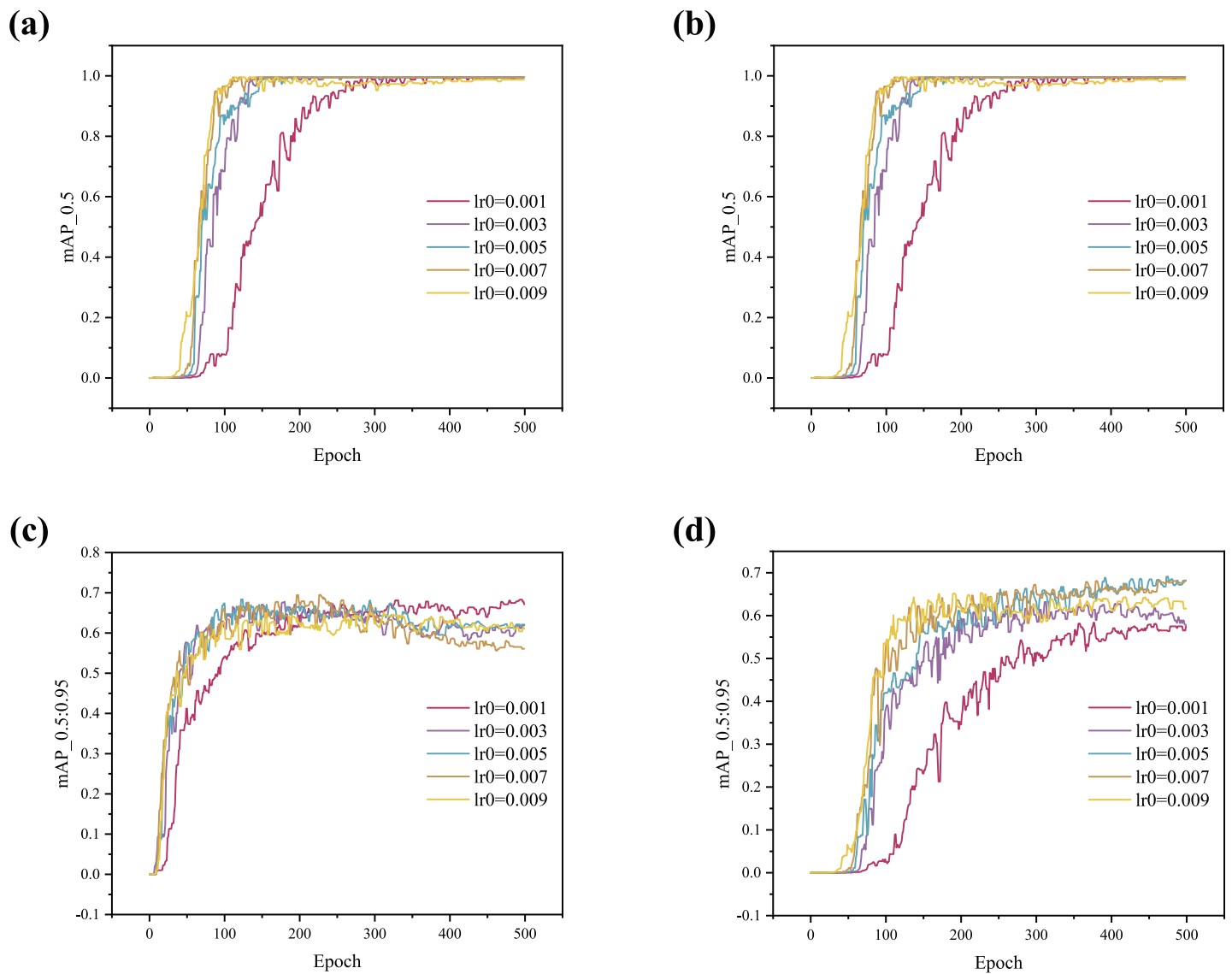

**Figure 4 (A–D) Training curve comparison between SGD and Lookahead with different lr0.**

| Table 3 | Ablation study of each component. | | |
|---|---|---|---|
| **Setting** | **FEM** | **FFM** | **mAP_0.5:0.95↑** |
| A1 | ✓ | | 0.63196 |
| A2 | ✓ | ✓ | 0.69141 |

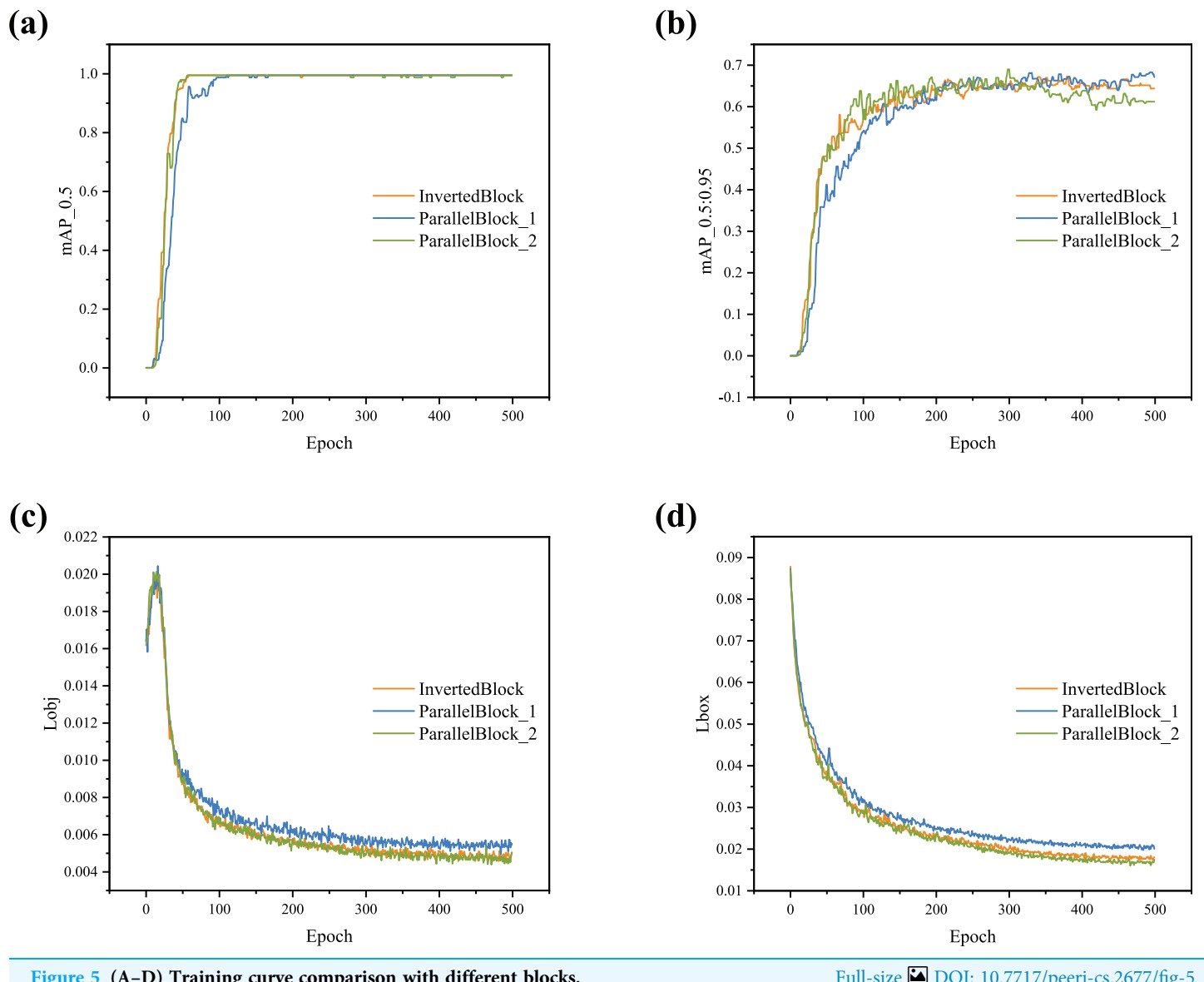

**Figure 5** (A–D) Training curve comparison with different blocks.               

when using the nearest neighbor method, the curves oscillate, fail to converge, and the detector's robustness is poor.

In summary, the detector using CARAFE outperforms the one using the nearest neighbor in terms of loss degree, mAP, Precision, and Recall, demonstrating stronger generalization ability. This experiment fully verifies the effectiveness of the proposed FFM.

## Comparison with other detection methods

In this section, to further analyze the performance of the proposed method, we compare it with other detection methods, such as YOLOv5s, YOLOv3 (*Redmon & Farhadi, 2018*), YOLOv5-RepVGG (*Ding et al., 2021*), and YOLOv5-Ghost (*Han et al., 2020*). For YOLOv5 and YOLOv3, we use the official open-source code provided. For YOLOv5-

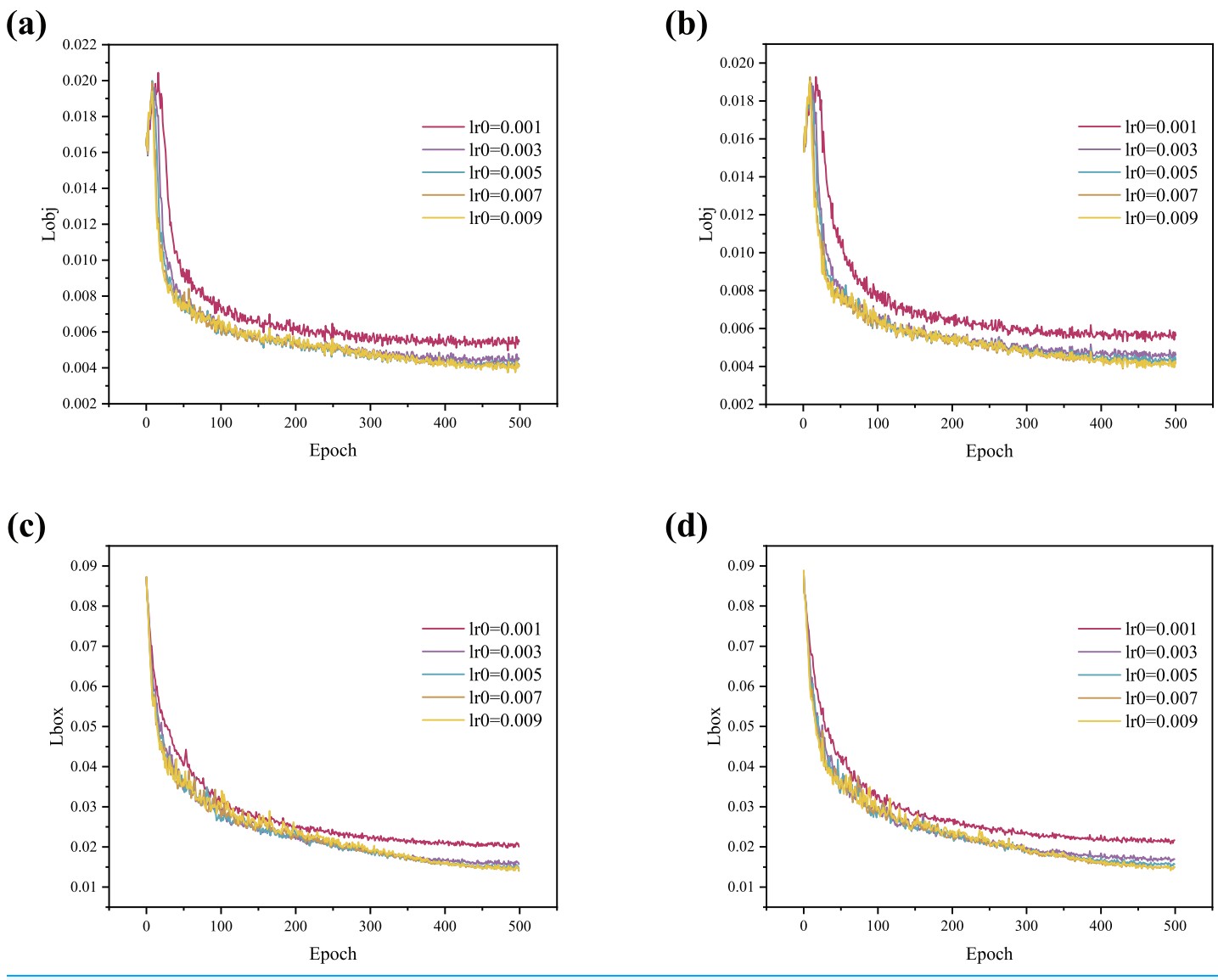

**Figure 6** (A–D) Loss curve comparison between nearest and CARAFE with different lr0.

RepVGG and YOLOv5-Ghost, we integrate the RepVGG Block and Ghost modules as the backbone, respectively, and test them under the same conditions. These models represent the optimal solutions from multiple experiments. The experimental results for the different methods on the coal-gangue dataset are shown in Table 4.

From an overall perspective, the proposed method demonstrates stable performance. Compared with YOLOv5s, YOLOv3, and YOLOv5-RepVGG, our method achieves a significant reduction in model volume by 80.29%, 97.71%, and 73.47%, respectively. Although YOLOv5-Ghost achieved better volume reduction, its mAP_0.5:0.95 is 12% lower than that of our method.

**Peer**J Computer Science

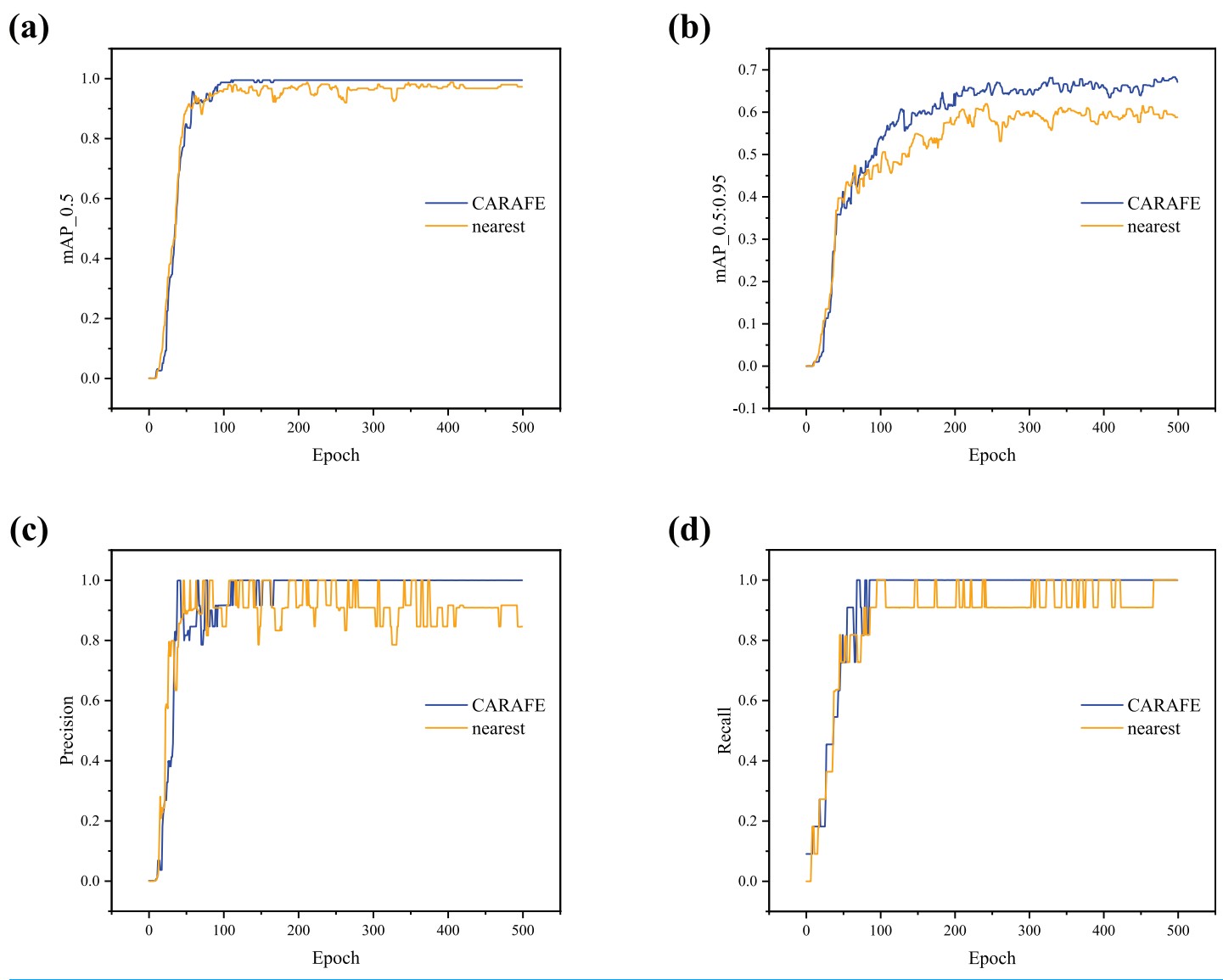

**Figure 7 (A–D) Training curve comparison with different upsampling methods.**

**Table 4 The experimental results of different methods on the coal-gangue dataset.**

| Method | mAP_0.5↑ | mAP_0.5:0.95↑ | Params↓ | GFLOPs↓ | Volume (M)↓ |
|---|---|---|---|---|---|
| YOLOv5s | 0.9950 | 0.6756 | 7,022,326 | 127.57 | 13.60 |
| YOLOv3 | 0.9950 | 0.6286 | 61,523,734 | 1,242.08 | 117.00 |
| YOLOv5-RepVGG | 0.9950 | 0.6114 | 5,189,502 | 74.30 | 10.10 |
| YOLOv5-Ghost | 0.9950 | 0.6064 | 5,088,078 | 85.82 | 2.41 |
| Ours | 0.9950 | 0.69141 | 986,758 | 67.67 | 2.68 |

An in-depth analysis reveals that the lower volume of the proposed method can be attributed to three main factors:

- First, the use of depthwise convolution, where the kernel performs feature extraction on a single channel, effectively reduces computational load.
- Second, the identity mapping operation allows the input of the convolution operation to be added directly to the output, without introducing additional parameters or computational complexity.
- Third, during upsampling, the lightweight operator compresses the feature map channels, further reducing the volume.

In terms of parameters and GFLOPs, the proposed method achieves a marked reduction in both metrics. Specifically, compared to YOLOv5s, YOLOv3, and YOLOv5-RepVGG, the number of Parameters is reduced by 85.95%, 98.40%, and 81.02%, respectively. Similarly, the GFLOPs are reduced by 46.95%, 94.56%, and 8.94%, respectively. These reductions underscore the efficiency of the proposed model, which significantly reduces computational complexity while maintaining high performance. This efficient model design is crucial for ensuring both practical feasibility and deployment cost-effectiveness, especially in resource-constrained environments.

In summary, the proposed method offers specific advantages: It effectively improves the detection performance for coal-gangue detection while addressing limitations in existing methods. By utilizing the feature extraction module (FEM) and the feature fusion module (FFM), the model not only enhances accuracy but also ensures low-cost deployment, thereby improving its potential for industrial application.

## CONCLUSION AND FUTURE WORK

In this article, we propose P-RNet for coal-gangue detection. The FEM is utilized to extract features of coal-gangue, addressing the complex background that complicates context information retrieval. FFM is introduced to improve prediction accuracy and improve robustness against complex backgrounds. Additionally, the Lookahead optimizer is employed to dynamically adjust the learning rate. Experiments on the coal-gangue dataset demonstrate that the proposed method achieves excellent performance and has low deployment costs, with each component (*i.e.*, FEM and FFM) proving to be practical. Overall, the proposed method achieves performance comparable to existing coal-gangue detectors and shows strong generalizability in coal gangue detection tasks.

In future work, we plan to explore two key directions. First, we aim to optimize the proposed algorithm to meet the standards of industrial production safety, advancing coal mine intelligence by focusing on enhancing real-time detection capabilities and minimizing error rates. Second, we intend to leverage generative adversarial networks (GANs) for image deblurring, which is anticipated to further improve the performance of the detector and support more robust decision-making in complex environments.

Additionally, while some components of the current work, such as the Lookahead optimizer, are well-established methods, our future efforts will emphasize unique

architectural improvements and advancements in self-supervised learning. These efforts aim to address the identified gaps and further enhance the novelty of the proposed approach.

### Funding
This work was supported by the Open Fund of Anhui Key Laboratory of Mine Intelligent Equipment and Technology (Grant No. ZKSYS202204). The funders had no role in study design, data collection and analysis, decision to publish, or preparation of the manuscript.

### Grant Disclosures
The following grant information was disclosed by the authors:
Open Fund of Anhui Key Laboratory of Mine Intelligent Equipment and Technology: ZKSYS202204.

### Competing Interests
The authors declare that they have no competing interests.

### Author Contributions

- Shexiang Jiang conceived and designed the experiments, performed the experiments, analyzed the data, authored or reviewed drafts of the article, and approved the final draft.
- Xinrui Zhou conceived and designed the experiments, performed the experiments, performed the computation work, prepared figures and/or tables, authored or reviewed drafts of the article, and approved the final draft.

### Data Availability
The data is available at GitHub and Zenodo:
- https://github.com/xinrui-z/-P-RNet
- Xinrui Zhou, & Jiang, S.-X. (2025). xinrui-z/-P-RNet: First release of my awesome software (v1.0.0). Zenodo. https://doi.org/10.5281/zenodo.14722446
The code is available in the Supplemental File.

### Supplemental Information
Supplemental information for this article can be found online at http://dx.doi.org/10.7717/peerj-cs.2677#supplemental-information.

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
