# Peer review of "A lightweight coal-gangue detection model based on parallel deep residual networks"

_PeerJ Computer Science, doi:10.7717/peerj-cs.2677_

## Round 0.1 · original submission · Major Revisions

Dear Authors,

Your paper has been revised. Based on the reviewers' reports, it needs major revision before being considered for publication in PEERJ Computer Science.
Please revise and resubmit your paper once you have addressed the concerns and criticisms of the reviewers. They have added good insight on improving your article further.

More precisely, the following points need to be clarified:
1) You must clearly explain the foundation of the P-RNet model described in your paper, defining the role of YOLO, ResNet, and MobileNet modules understandably;
2) You must better elucidate the limits on the P-RNet and its computational requirements.

Reviewer 1 ·

Basic reporting

Clear and unambiguous, professional English used throughout.: Yes

Literature references, sufficient field background/context provided.: Yes

Professional article structure, figures, tables. Raw data shared.: Yes

Self-contained with relevant results to hypotheses.: Yes

Formal results should include clear definitions of all terms and theorems, and detailed proofs.: Yes

Experimental design

Original primary research within Aims and Scope of the journal.: Yes

Research question well defined, relevant & meaningful. It is stated how research fills an identified knowledge gap.: Yes

Rigorous investigation performed to a high technical & ethical standard.: Yes

Methods described with sufficient detail & information to replicate.: Yes

Validity of the findings

Impact and novelty not assessed. Meaningful replication encouraged where rationale & benefit to literature is clearly stated.: Yes

All underlying data have been provided; they are robust, statistically sound, & controlled.: Yes

Conclusions are well stated, linked to original research question & limited to supporting results.: Yes

Additional comments

Dear authors,

I have now completed the review of the manuscript titled "A lightweight coal-gangue detection model based on parallel deep residual networks."

In the present study, authors propose a new lightweight model called P-RNet that combines parallel deep residual networks with feature fusion techniques. This appears to be an innovative architectural design for this specific application.

The authors conduct extensive ablation studies and comparisons with other state-of-the-art models to validate their approach. This helps demonstrate the effectiveness of different components. The results show P-RNet achieves comparable or better accuracy (mAP) compared to larger models while significantly reducing model size, which is valuable for practical deployment.

The research addresses a real-world problem in coal processing and aims to improve automation and efficiency in this domain. Moreover, the paper provides a thorough explanation of the model architecture, training process, and design choices.

The manuscript is interesting and, in general, fairly well-written.

I have some suggestions to further improve the quality of the manuscript.

I would like to suggest that the authors address these limitations in the article, either by discussing them in the limitations section or, where feasible, by making the appropriate revisions:

Major:

1. The paper focuses mostly on accuracy metrics but doesn't delve deeply into where and why the model makes mistakes. This could provide valuable insights for further improvements. To improve this, I would like to recommend authors to compare similar SOTA articles that are not related to the coal-gangue detection.

2. While the model shows good performance on the test set, there's no evaluation on truly unseen data from different plants or conditions. This would better demonstrate practical applicability. The comparison models (YOLOv5s, YOLOv3, etc.) are not necessarily optimized for this specific task. A fairer comparison would be against models specifically designed or fine-tuned for coal-gangue detection.

3. While model size is reduced, there's little discussion of inference speed or computational requirements, which are crucial for real-time industrial applications. Moreover, Some components like the Lookahead optimizer are not novel to this work. More emphasis could be placed on what is truly unique about the proposed architecture. To improve this, I would like to suggest authors to discuss further research proposed in future. For example, Consequential Advancements of Self-Supervised Learning (SSL) in Deep Learning Contexts, etc...

Minor:

4. The paper doesn't mention if the code will be open-sourced, which limits reproducibility and potential impact on the field.

5. Given the industrial context, there's limited discussion on the safety implications of false positives/negatives or how the system would integrate with existing safety protocols.


Thank you for your valuable contributions to our field of research. I look forward to receiving the revised manuscript.

·

Basic reporting

This paper proposes a coal-gangue detection model that uses parallel deep residual networks. One of the key contributions highlighted in the title is the “lightweight” nature of the model. However, I could not find any data or evidence supporting its lightweight characteristics. If the authors propose a lightweight model, they should provide the parameters and FLOPs (Floating Point Operations Per Second) to substantiate their claims.

Additionally, I am confused about the type of basic model used in the paper. Is it a classification model such as ResNet or MobileNet, or an object detection model like YOLO, or an RNN?

When I read subsection 2.1:
* * *
Inspired by deep learning methods (e.g., YOLO, ResNet, MobileNet, etc.) Sandler et al. (2018); He et al.84 (2015), a lightweight coal-gangue detection model named P-RNet is designed, which is based on a parallel deep residual network. The overall architecture of P-RNet is illustrated in Figure 1, and it mainly consists of four parts: Input, Backbone, Head, and Output.
* * *
It is challenging to understand the foundation of the model described in the paper. YOLO is an object detection model, ResNet is a CNN-based classification model, and MobileNet is a lightweight CNN-based classification model. I am confused about which of these models or components is being used as the basis for the P-RNet model.

Figure 1 shows the proposed P-RNet architecture, which appears to have a structure similar to YOLO, with a Backbone, Head, and Neck. If this is the case, I would recommend providing a more detailed explanation to clarify how the model builds upon or integrates elements from YOLO. If not, please explain how a CNN-based classification model (such as ResNet or MobileNet) can be adapted for object detection in this context.

Furthermore, I suggest including object images as examples to better illustrate the model’s real-world application and make the concept easier to understand."

Experimental design

As above

Validity of the findings

NON

Additional comments

NON

Reviewer 3 ·

Basic reporting

Overall the topic of the paper is good.

Experimental design

Experimental design is overall good however some explanations are required.

Validity of the findings

Improvements required are detailed in detailed list of comments.

Additional comments

This paper presents a forgery detection technique titled, “A Lightweight Coal-Gangue Detection Model Based on Parallel Deep Residual Networks”. The topic is good however I have some points that need to be addressed.
My detailed concerns on the study are:
1. The study presents that complex condition in actual mining environment or plant, however I more eager to know if the model’s performance is evaluated in real-world complex conditions or not?
2. Section 2.3 presents feature fusion which make feature length larger. The process increases the computational complexity however the paper does not address it.
3. The model using in the presented study is trained on small and the only dataset from coal preparation plant in Anhui, China. What will be the generalizability of the presented study in other environments or plants having different conditions?
4. As the study presents that the model is lightweight, then how this will be deployed in low-power or low resource devices on coal sites.
5. The study is silent for diverse conditions in real-time environment such as dust which may affect the performance of the proposed model.
6. In continuation of the comment 5 why noise reduction or contrast enhancement is not considered along with linear transformation and deblurring?
7. The ParallelBlock and CARAFE features are used, more detail is required as why these are selected and fused, what benefit they have?
8. The study lack proper detail on how the model will distinguish coal-gangue in coal.
9. The study lack how the Lookahead Optimizer improve convergence.
10. Figure 4-7 are not legible, the authors should provide high quality images.

---

## Round 0.2 · accepted · Accept

Dear Authors,

Your manuscript has been accepted for publication in PEERJ Computer Science. Thank you for your fine contribution.

Reviewer 1 ·

Basic reporting

All comments have been thoroughly addressed. I extend my gratitude to both the authors and editors for taking my opinions into consideration during the review of this manuscript.

Experimental design

No further issues.

Validity of the findings

No further issues.